# Surface uplift in the Central Andes driven by growth of the Altiplano Puna Magma Body

Jonathan P. Perkins[1,†], Kevin M. Ward[2], Shanaka L. de Silva[3], George Zandt[2], Susan L. Beck[2] & Noah J. Finnegan[1]

The Altiplano-Puna Magma Body (APMB) in the Central Andes is the largest imaged magma reservoir on Earth, and is located within the second highest orogenic plateau on Earth, the Altiplano-Puna. Although the APMB is a first-order geologic feature similar to the Sierra Nevada batholith, its role in the surface uplift history of the Central Andes remains uncertain. Here we show that a long-wavelength topographic dome overlies the seismically measured extent of the APMB, and gravity data suggest that the uplift is isostatically compensated. Isostatic modelling of the magmatic contribution to dome growth yields melt volumes comparable to those estimated from tomography, and suggests that the APMB growth rate exceeds the peak Cretaceous magmatic flare-up in the Sierran batholith. Our analysis reveals that magmatic addition may provide a contribution to surface uplift on par with lithospheric removal, and illustrates that surface topography may help constrain the magnitude of pluton-scale melt production.

[1] Department of Earth and Planetary Sciences, University of California Santa Cruz, 1156 High Street, Santa Cruz, California 95064, USA. [2] Department of Geosciences, The University of Arizona, 1040 E. 4th Street, Tucson, Arizona 85712, USA. [3] College of Earth, Ocean, and Atmospheric Sciences, Oregon State University, 104 CEOAS Administration Building, Corvallis, Oregon 97331-5503, USA. † Present address: U.S. Geological Survey, 345 Middlefield Road, Menlo Park, California 94025, USA. Correspondence and requests for materials should be addressed to J.P.P. (email: jperkins@usgs.gov).

The Altiplano-Puna Magma Body (APMB) resides within the Altiplano-Puna plateau, a region that is characterized by a higher mean elevation[1,2] than the Altiplano to the north, a thickened crust[3,4], and a very thin mantle lithosphere[4]. The thin lithosphere beneath the APVC may result from convective removal[5–7], a potentially cyclical process[8] that has contributed to pulses of rapid surface uplift throughout the history of the Central Andes[9]. The mantle heat flux associated with lithospheric removal is thought to be responsible for the flare-up of large-volume ignimbrites[5,6,10], collectively known as the Altiplano-Puna Volcanic Complex (APVC)[11], since ~11 million years ago (Ma)[12]. Chemical compositions of the ignimbrites show a roughly equal contribution from crustal and mantle sources[11,13], consistent with extensive melting of the upper mantle and crust. Though horizontal shortening alone may not account for the crustal thickness observed in the APVC[1], the role of magmatic addition in contributing to the crustal evolution and surface uplift history of the region is often neglected, largely due to the lack of constraints on the volume of magmatic material within the crust[1,2]. Although the presence of substantial melt beneath the APVC has previously been recognized[14], new high-resolution tomography by Ward et al.[15] reveals an immense, ~500,000 km$^3$ zone of partial melt at 10 to 20 km depth beneath the APVC, which substantially changes predictions for both the APMB's magma production rate and contribution to crustal thickening (and thus surface uplift).

The primary goal of this paper is to test whether melt intrusion into the APMB is actually reflected in the topography of the APVC, and if so, utilize that topographic signature to help place bounds on estimates of mantle melt intrusion into the crust. Our analysis that follows is comprised of the following sections. We first examine anomalies in the long-wavelength topography of the Central Andes, and determine whether or not observed anomalies are structural and therefore recorded in the topography of the underlying basement rocks of the APVC, or are merely the result of ignimbrite deposition on the plateau. We then analyse free-air, Bouguer and isostatic residual gravity anomalies and examine to what degree the topography of the APVC appears isostatically compensated. Pairing topographic data with geochemical constraints on the magmatic system of the APVC, we estimate the volume of the APMB with an isostatic magma production rate model. Finally, we attempt to constrain surface uplift rates of the APVC over its lifetime from 11 Ma using the volcanic record as a proxy for magmatic thickening over million-year long timescales relevant to mantle melt flux[16], and place our findings in the context of the known tectonic and geodynamic evolution of the Central Andes between 20º and 25º S.

Through our topographic analysis, we find a tight spatial correlation between a ~1 km high, long-wavelength topographic dome and the zone of low seismic velocities that is thought to characterize the APMB. The dome appears to be a structural, rather than depositional feature, and gravity data are broadly consistent with Airy isostatic compensation of the topographic anomaly. APMB volumes and magma production rates modelled from its topographic signature compare well with independent estimates from Ambient Noise Tomography[15], and our surface uplift rate calculations suggest that magmatic flare-ups can grow topography at rates similar to uplift from lithospheric removal.

## Results

**Topography of the APMB.** To constrain the topographic signal associated with the presence of the APMB (Fig. 1, Supplementary Fig. 1), we measure the long wavelength component of the topography in the Central Andes and find that a high-amplitude, kilometre-scale dome spatially coincides with the estimated

bounds of the seismically imaged magma body (Fig. 1b and 2a). Here the seismic bounds of the APMB in Figs 1b are delineated by the 2.9 km s$^{-1}$ contour from the shear-wave velocity model of Ward et al.[15] (Figs 1b and 2d). The dome also correlates with the extent of ignimbrites and concentration of calderas within the APVC (dashed yellow line and black circles in Fig. 1a), as well as a low Bouguer gravity anomaly[3] (dashed black line, Fig. 1b).

Calculating the true amplitude of the long-wavelength topographic dome is complicated because the topography is not symmetric on either side of the dome (Fig. 2a). To the north, the mean elevation of the topography lowers as the plateau transitions to the low-relief Altiplano surface. To the south of the APVC lies the Puna plateau, which is characterized largely by high-relief, reverse fault-bounded blocks[2]. From the profile of the lowpass-filtered topography, we estimate an amplitude range from 900 to 1,400 m. The middle of this range is 1,100–1,200 m, which is approximately equal to the magnitude of structural relief estimated from tilted forearc basin strata for this section of the Central Andes since 11 Ma by Jordan et al.[17].

**Analysis of basement rock structure.** A key question is whether the high topography of the APVC results from structural uplift, or is merely the result of volcanic material deposited on the plateau surface[18]. We attempt to address this issue both by estimating the mean thickness of ignimbrite deposits along the plateau, and by measuring the elevations of basement rock outcrops in cross-sections through our topographic anomaly.

We estimate the mean thickness of ignimbrite material above the APMB basement using ignimbrite volume estimates of Salisbury et al.[12] who report a total dense rock equivalent volume of 12,800 km$^3$. Over the ~55,000 km$^2$ area of the APVC, roughly 20% consists of caldera basins where much of this material resides. Assuming an average caldera fill depth (defined as the relief from resurgent dome top to caldera moat) of 710 m (ref. 12), we estimate an average thickness of ~110 m across the non-caldera basins (Supplementary Table 1).

To verify this estimate, we map exposures of basement rocks proximal to our longitudinal and latitudinal cross section lines (Fig. 1b). The basement rocks underlying the late Neogene ignimbrites within the Central Andes are largely composed of the Paleozoic Antofalla terrane, an accreted crustal block consisting of metamorphosed igneous and sedimentary basin rocks associated with the Sunsás orogeny[19]. Figures 2 and 3 show the mean elevation of the mapped basement outcrops projected onto our cross section lines. Although exposures of basement rock become less common toward the interior of the APVC, the median outcrop elevations roughly track the rise observed in the topographic data sets and appear to define the level of topography just below the volcanoes (red lines, Figs 2a and 3a). Last, evidence for a ~1 km structural rise is also expressed along the monoclinal folds of the western slope of the Central Andes, where Jordan et al.[17] find an increase in structural relief growth of ~1.1 km from 20º to 24º S since 11 Ma.

**Gravity estimates of isostatic equilibrium.** The dome is also located within the bounds of a < −300 (ref. 20) to −400 (ref. 3) mGal Bouguer gravity anomaly, which roughly mirrors topography and suggests a thickened[21], possibly low density[3] crust directly below the dome (Figs 1b and 2b). A high-resolution crustal thickness model does not exist for the lower crust directly beneath the APMB, making direct comparison with crustal thickness difficult. However, it is not clear that a seismic Moho would even be easy to detect within an active magmatic zone, as the dense cumulate layering and mafic melt interaction within the lower crust will produce a complex and thick transition zone

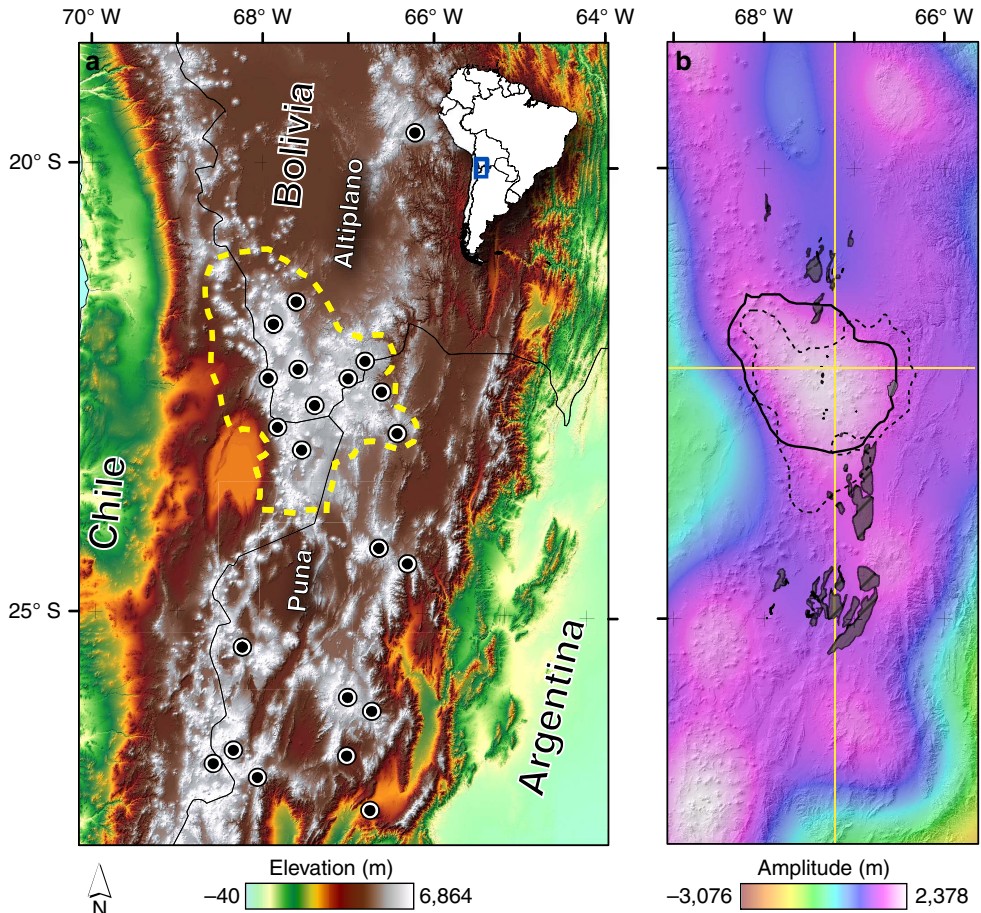

**Figure 1 | Location map of the study area in the Central Andes.** (**a**) The approximate extent of the Altiplano-Puna Volcanic Complex (APVC; dashed yellow line) is seen. Caldera locations are denoted by black circles. (**b**) Shows the locations of basement outcrops (grey shaded areas), the 2.9 km s$^{-1}$ velocity contour that roughly defines the extent of the APMB (solid black line) and the −400 mGal Bouguer gravity anomaly of *Prezzi et al.*[3] (dashed black line) overlain on the long wavelength topography. Longitudinal and latitudinal cross section lines for Figs 2 and 3 are shown in yellow.

of seismic velocities that may obfuscate the Moho boundary[22]. Regardless, Airy isostatic residual anomalies from the World Gravity Model 2012 (ref. 20) are roughly zero at the APVC (Fig. 3), consistent with isostatically supported topography.

In addition, although there is a nonzero free-air gravity anomaly along the Central Andes (Figs 2 and 3), it is low and does not change markedly from the Altiplano to the Puna (Fig. 2c), which further suggests that the Central Andes as a whole are roughly in isostatic balance. Here the free-air gravity anomaly's positive value likely arises from the narrow width of the Central Andes (∼200 km) relative to the compensation depth (∼60 km), which may result in significant free-air edge effects within the APVC[23]. In addition, short wavelength (and thus flexurally supported) topography such as volcanoes appears to contribute to this signal (Fig. 2b).

**Modelling melt production from topographic data.** The presence of the large-volume APMB implies a substantial input of mantle-derived magma into the crust of the Altiplano Puna Volcanic Complex. As the regional gravity data suggest that the topography of the APVC is in isostatic equilibrium, we can therefore exploit the topographic high above the APMB to learn about the contribution of magmatic thickening to surface uplift. To do so, we invert the topographic profile of the APMB dome ($H_b$) to solve for the initial load added to the base of the crust ($W_i$) using a buried load isostatic model (Equation 5; refs 24,25,

Fig. 4; see the 'Methods' section for our model description). We then calculate the magmatic contribution to crustal loading using the arc mantle magma production rate model of *Ward et al.*[15] (Fig. 5). This allows us to estimate the volume of the APMB directly from the topography, and provides an independent value to compare with APMB volume estimates from seismic tomography[15].

Our isostatic model requires a constant crustal density, so we take an average density of 2,800 kg m$^{-3}$ for the crustal column and 3,250 kg m$^{-3}$ for the density of the underlying asthenosphere[3]. This model therefore precludes examining the effect of crustal thickening on mineralogical (and thus density) changes to lower crust over time like the growth of garnet pyroxenite[26,27]. Given that the growth of the AMPB is thought to occur after convective removal of dense lithospheric mantle and lower crust[8,28,29], neglecting such density changes from metamorphic reactions in this instance seems appropriate. It is worth noting, however, that if magmatic thickening from growth of the APMB has indeed triggered the growth of garnet pyroxenite in the lower crust, then the subdued topography from the dampened isostatic response would lead to an under-prediction of melt volume. Furthermore, here we do not consider the effects of erosion on a magmatically thickened crust[27]. Precipitation rates drop significantly from the eastern flank of the Central Andes to the plateau surface[30], and the high preservation of ignimbrites that blanket the landscape of the Altiplano-Puna plateau suggest minimal erosion over the

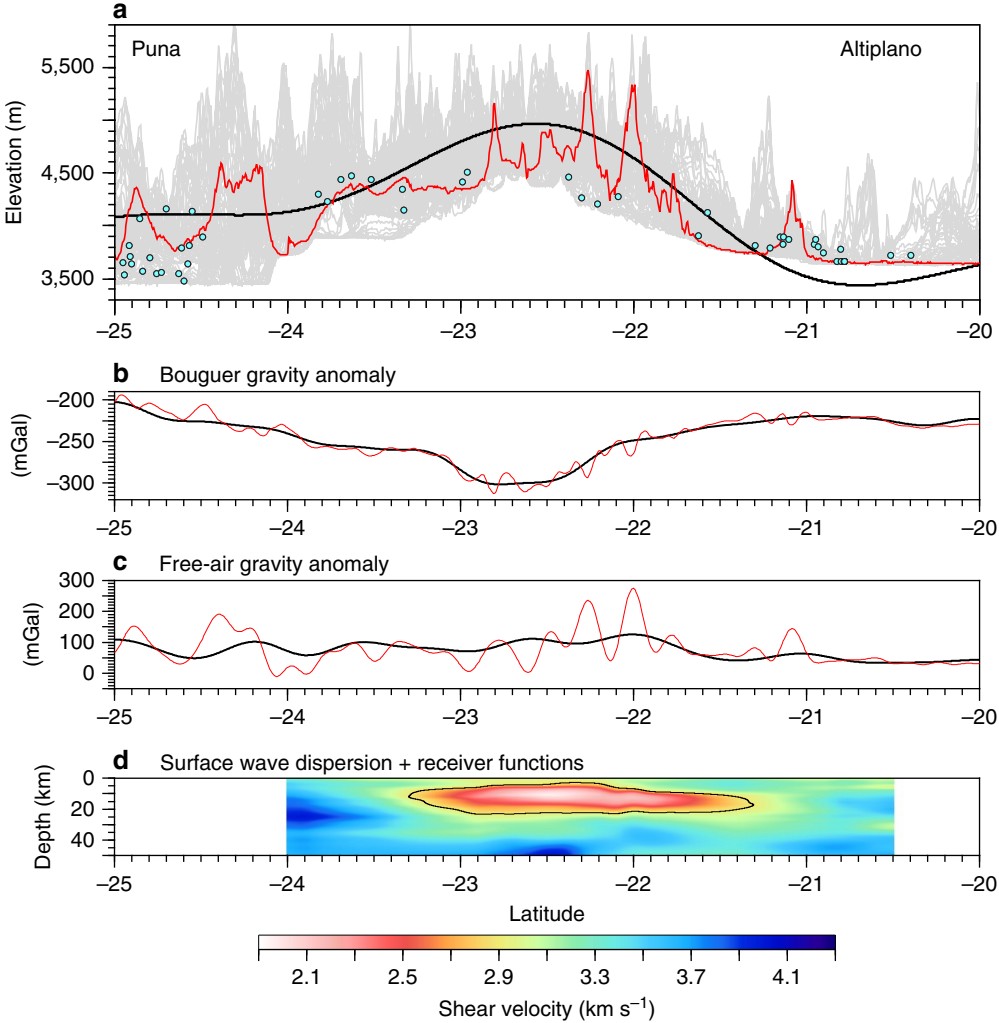

**Figure 2 | Longitudinal cross section of topographic and geophysical datasets.** (**a**) Shows a topographic swath profile (grey band) along the 67.2° longitude line, an exact topographic profile along the cross-section line (red line), filtered long wavelength topography (black line) and median basement rock outcrop elevations (blue dots). (**b**) Shows the filtered (black) and unfiltered (red) Bouguer gravity anomaly from the AnGrav data set. (**c**) Shows the filtered (black) and unfiltered (red) free-air gravity anomaly from the AnGrav data set. (**d**) Shows the S wave velocity models of Ward *et al.*[15]. Black line represents the 2.9 km s$^{-1}$ velocity contour that corresponds to the approximate boundary of the partial melt.

course of the 11 Myr ignimbrite flare-up. Although erosion rates estimated from the incision of variably-aged volcanic flanks in the APVC are ∼7–9 m Ma$^{-1}$ (ref. 31), sufficient to erode through the thickness of ignimbrite deposits over the lifespan of the flare-up, these rates are likely significantly less on the flat valley floors where ignimbrite deposits lie. Therefore, we do not consider these processes in our isostatic model.

The arc mantle magma production rate is modelled using a geochemical mass balance, and requires knowledge of both the ratio of crustal to mantle provenance ($\mu$) and the ratio of dense residue to melt mass ($\eta$) within the batholith (Equation 4). Isotopic modelling using the chemistry of erupted APVC ignimbrites[13], as well as geochemical investigations of batholiths and their lower crustal cumulates[32], suggests that both $\mu$ and $\eta$ are approximately ∼1:1 for the APVC[2] (see the 'Methods' section for full description). Effectively, this implies that the APMB volume should be approximately equal to the magnitude of the modelled isostatic load volume $W_i$ (Fig. 5).

From our isostatic melt production model, we calculate an APMB volume of ∼480,000 km$^3$ (Table 1, Supplementary Fig. 2). Our estimate calculated using the topography above the APMB is close to the seismically determined magma chamber volume of

∼530,000 km$^3$ from the 2.9 km s$^{-1}$ velocity contour of Ward *et al.*[15]; however, our calculation is sensitive to both the prescribed density difference between the crust and mantle asthenosphere, as well as the topographic amplitude of the dome used in our isostatic model, so we include estimates from a range of parameter values in Supplementary Table 2 and Supplementary Fig. 2. Taking into account both the seismic and topographic constraints on mantle melt flux, we calculate a plutonic:volcanic ratio ($\beta$) of 32 and an arc mantle magma production rate ($\xi$) of ∼195 km$^3$ km$^{-1}$ per million years for the APMB (Equation 5, Table 1; see the 'Methods' section for details).

## Discussion

The APMB cross-sectional growth rate (∼200 km$^3$ km$^{-1}$ per million years) appears to exceed the peak magmatic addition rate of the late Cretaceous magmatic flare-up in the Sierra Nevada batholith (SNB) system. Estimates of magmatic addition rates for the SNB range from 63 km$^3$ km$^{-1}$ per million years considering the upper 30 km of crust[33] to 144 km$^3$ km$^{-1}$ per million years considering the upper 70 km of crust[33]. Given that our model considers only the production of mafic cumulates during

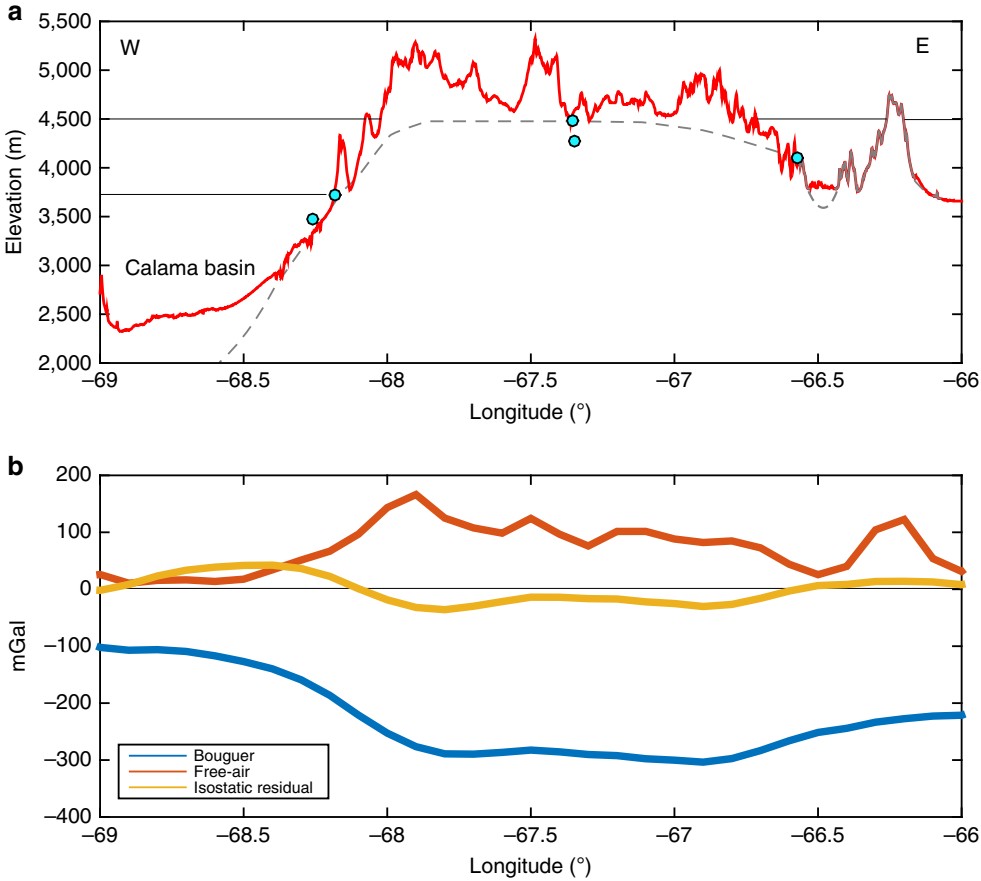

**Figure 3 | Latitudinal cross section of topographic and geophysical datasets.** (**a**) Shows a topographic profile (red line) along the 22.4° latitude line and isolated basement rock outcrop elevations (teal dots, locations in Fig. 1b). The grey line represents a coarse estimation of the elevation of pre-11 Ma basement rock. It is solid where it is visible at the surface in the Eastern Cordillera, and is dashed underneath the APVC and the Calama basin to the west. Reference lines are at 3,700 m (mean elevation of the southern Altiplano) and 4,500 m (mean elevation of the Altiplano-Puna). (**b**) Shows gravity anomalies along the cross section line. The free-air gravity anomaly mirrors the flexurally supported short-wavelength topography, and the negative Bouguer gravity anomaly roughly mirrors the long-wavelength topography. The lack of a significant Airy isostatic residual anomaly suggested that the topography of the APVC is isostatically compensated.

batholith growth and does not consider partial melting of the lower crust and restite production directly (which likely contributes to the higher magmatic addition rate for the 70 km depth value), a more appropriate comparison for batholith growth between the APMB and SNB is the 30 km depth value of 63 $km^3 km^{-1}$ per million years. This implies that the APMB flare-up is outpacing the peak SNB flare-up by over a factor of two, and is closer to the magmatic addition rates associated with intra-oceanic volcanic arcs and spreading ridges[34]. Our topographically constrained estimate of $\beta = 32$ for the plutonic:volcanic ratio of the APMB, however, is approximately equal to the geochemical estimates of $\beta = 30$ for the Sierran batholith[33]. Although the sill-like radial geometry of the APMB differs from the elongate geometry of the Sierran batholith, magmatism during the peak APVC flare-up focused within a relatively narrow NNW-SSE zone akin to most arc-related batholiths[35,36]. Furthermore, the APVC is only one component of a much more extensive ignimbrite province that spans the Central Andean arc and therefore may be considered analogously to the SNB[37,38].

Studies of large silicic magma systems like the APVC show evidence for substantial thermal[35], rheological[10] and mineralogical[22] modification of the crustal column arising from melt generation. These changes can potentially have profound effects on the rates and style of tectonics in the growth of a continental arc. For example, Victor *et al.*[39] show that the

timespan of E-W shortening along the west Altiplano fault system at the outer western fringes of the Andean slope coincides with the occurrence of ignimbrite volcanism across the span of the Central Andes, and argue that thermal weakening of the crust from intense magmatism should promote increased shortening rates (for example, ref. 10). Farther east along the Western Andean Slope, the onset of long-wavelength monoclinal folding at 10 Ma (ref. 17) coincides with the onset of the APVC ignimbrite flare-up, and an excess structural relief of 1.1 km is seen in tilted forearc basin strata at these latitudes[17]. Large-scale viscous warping of the plateau flank is consistent with deformation of thermally weakened crust, and both modelling[10] and myriad observational data[40–42] suggest that viscous processes are likely occurring throughout the crustal column of the APVC.

The ~1 km of surface uplift gained over the 11 million years of magmatic addition represents approximately one-fifth of the total modern-day elevation of the Altiplano-Puna, with the remainder of surface uplift being accomplished primarily through crustal shortening[1,2] and lithospheric removal[5]. Given the volcanological evidence for episodic magmatism throughout the APVC flare-up[35], it is likely that surface uplift from magmatic addition was non-uniform over time. As the peak ~5 million year flare-up timescale potentially records the time variation of mantle power input into the crust[16], we use the record of volcanic volume estimates[12] as a proxy for plutonic magmatic flux and estimate isostatic uplift rates over the peak volcanic pulse of the

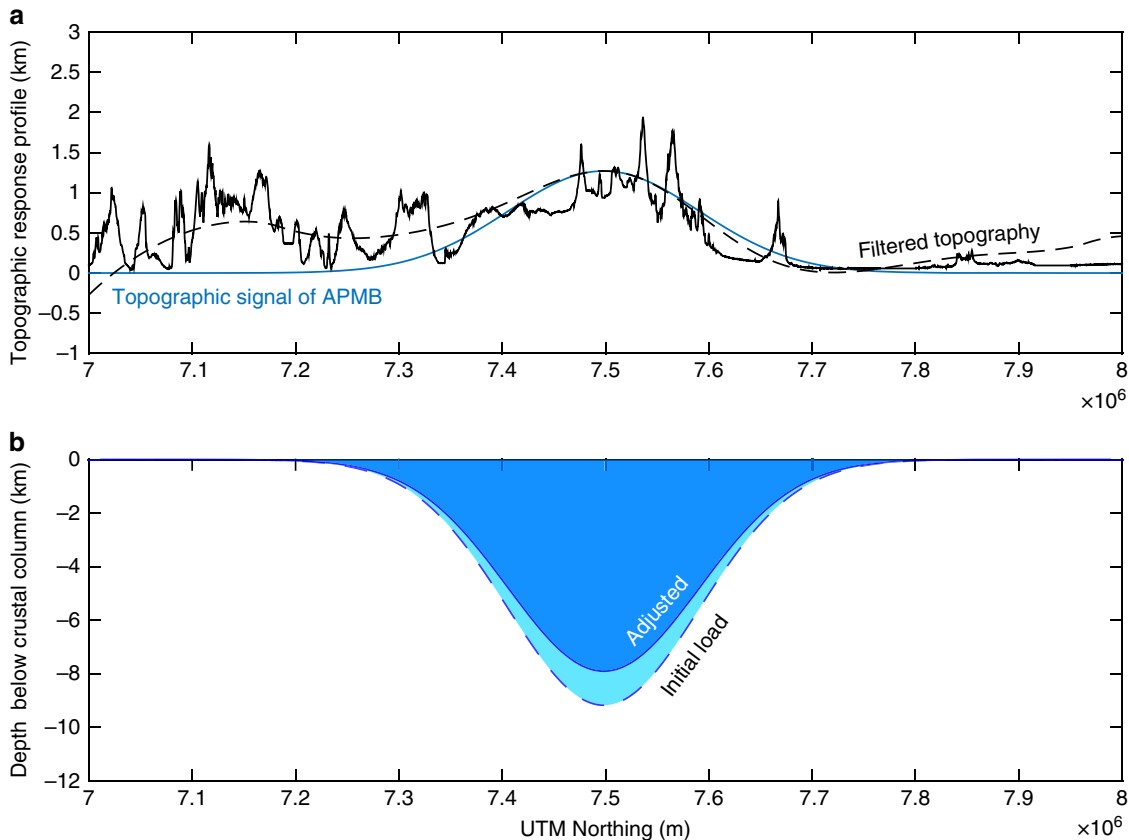

**Figure 4 | Buried load isostatic model results for the topographic dome overlying the Altiplano-Puna Magma Body.** (**a**) Shows the north–south raw topographic profile using 90 m SRTM data (solid black line), the long-wavelength component of that profile (dashed black line, 175 km cutoff wavelength), and the Gaussian approximation of the topography related to the APMB (solid blue line). (**b**) Shows the isostatic model profiles of the initial (light blue) and adjusted (darker blue) thickened crust beneath the APMB.

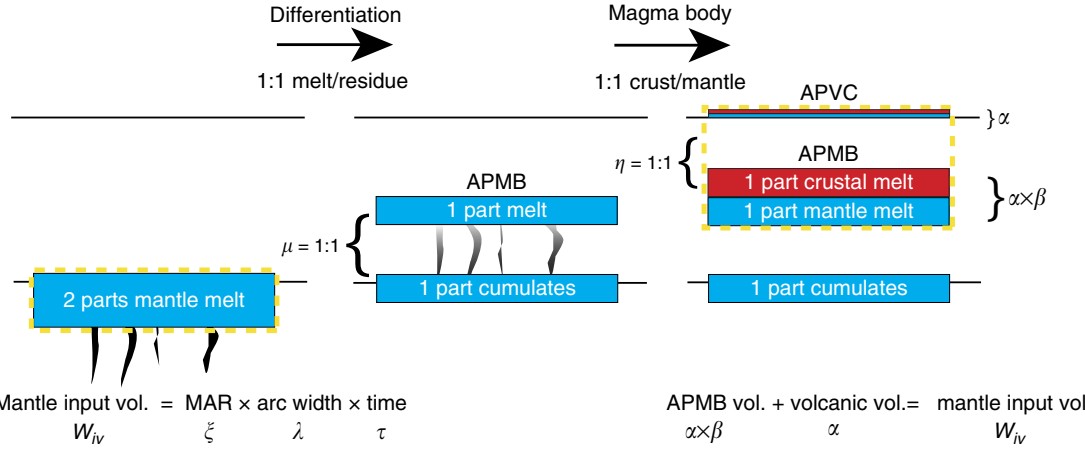

**Figure 5 | Schematic diagram illustrating the geochemical mass balance of the arc mantle magma production rate model.** Here we show the case where both the ratio of upwardly ascending mantle melt to dense residue ($\mu$) and the crust to mantle composition of the melt body ($\eta$) are 1:1 (refs 6,13,32). In this circumstance, half of the total mantle melt ascends to the mid-crust and leaves the remainder as residue. The mid-crustal melt body then doubles in mass as it melts and assimilates the crust around it, making a 1:1 mixture of crust:mantle composition magma that has been documented in ignimbrites of the Altiplano Puna Volcanic Complex[12]. The Altiplano Puna Magma Body volume and its volcanic output are therefore directly comparable to the volume of mantle melt input. The dashed yellow boxes enclose equal volumes of material according to the mass balance argument.

APVC (Fig. 6). Estimated surface uplift velocity over the flare-up is approximately 0.2 mm per year, approaching the 0.4 mm per year rates associated with lithospheric removal in the southern Altiplano from 16 to 9 Ma (refs 9,43).

Although beyond the scope of this study, it may be possible to look for evidence of possible feedbacks between magmatically driven surface uplift and erosion in the APVC by using the variably aged ignimbrites as markers to measure

**Table 1 | Magma production rate calculations from isostatic model runs.**

| $H_b$ | $\rho_c$ | $\rho_a$ | $\Delta\rho$ | $W_{iv}$ | $\xi_{Wi}$ | $\beta$ | $V_{apmb}$ |
|---|---|---|---|---|---|---|---|
| (m) | (kg m$^{-3}$) | (kg m$^{-3}$) | (kg m$^{-3}$) | (km$^3$) | (km$^3$ km$^{-1}$ per million years) | | (km$^3$) |
| 800 | 2,800 | 3,300 | 500 | 287,610 | 114 | 18 | 272,610 |
| 800 | 2,800 | 3,250 | 450 | 314,730 | 124 | 20 | 299,730 |
| 800 | 2,800 | 3,200 | 400 | 348,620 | 138 | 22 | 333,620 |
| 1,000 | 2,800 | 3,300 | 500 | 358,580 | 142 | 23 | 343,580 |
| 1,000 | 2,800 | 3,250 | 450 | 392,380 | 155 | 25 | 377,380 |
| 1,000 | 2,800 | 3,200 | 400 | 434,640 | 172 | 28 | 419,640 |
| 1,200 | 2,800 | 3,300 | 500 | 450,830 | 178 | 29 | 435,830 |
| 1,200* | 2,800* | 3,250* | 450* | 493,330* | 195* | 32* | 478,330* |
| 1,200 | 2,800 | 3,200 | 400 | 563,540 | 223 | 37 | 548,540 |

*Preferred run given knowledge of density variability and simplest topographic characterization of long wavelength dome.

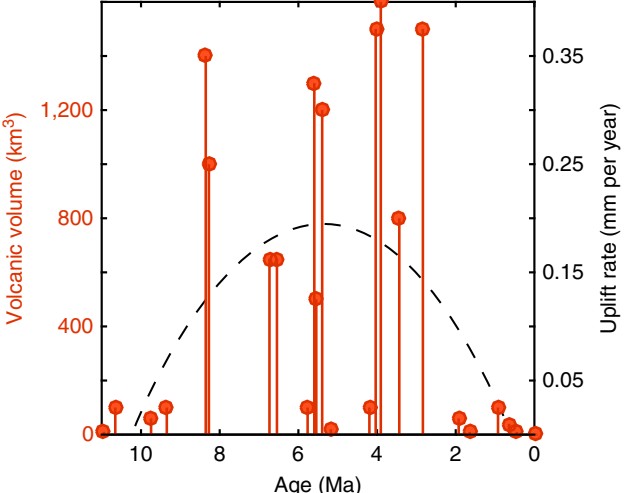

**Figure 6 | A model for the contribution to surface uplift from magmatic thickening of the crust since the onset of ignimbrite volcanism at 11 Ma.** Orange bars with circles show the erupted volumes of APVC ignimbrites (dense rock equivalent) since 11 Myr (ref. 12). The signal is dominated by an extended pulse of volcanism from 8 to 3 Myr, and this 5 Myr timescale likely correlates with variability in mantle power input[16] and thus magmatic thickening of the crust. We convert the volcanic output to its plutonic equivalent using our estimated value of $\beta$, and model the isostatic response to melt intrusion over this timescale. The dashed line represents a cartoon trajectory of surface uplift through the maximum value calculated over the 5 Myr flare-up.

changes in erosion rates since 11 Ma. To the extent that river incision rate is proportional to channel slope[44], one may be able to estimate an expected change in erosion rate over time by calculating a baseline change in channel slope. An elevation gain of 1.2 km over the 230 km half-width of the APMB dome yields a total change in long-wavelength surface slope of 0.005. Near Uturuncu volcano on the northern flanks of the long-wavelength dome in southern Bolivia, the slopes of rivers are presently ~0.01 (ref. 45), so the change in slope from magmatic tilting could potentially result in a doubling of erosion rate (assuming a linear scaling of erosion rates with slope[44]). Averaged over the 5 million year peak magmatic flare-up of the APVC, this would correspond to a slope change of ~0.001 (0.1%) per million years. This very slow forcing may prove difficult to detect in the geomorphic or sedimentary record, however, and is further complicated by the fact that erosion rates are expected to be highly heterogeneous as rivers in the APVC are still transiently

adjusting to the emplacement of ignimbrites and volcanoes on the surface[45,46].

In our modelling, we have also neglected to consider the feedbacks between magmatically driven surface uplift on surface weathering, erosion, and climate (for example, ref. 27). For example, the work of Lee et al.[27] suggests that enhanced $CO_2$ output from flare-ups of continental arc magmatism may contribute to greenhouse conditions, and the subsequent weathering and increased erosion of remnant arc high topography may lead to icehouse conditions over the 50 million years-timescale oscillations in Earth's climate[47]. In this instance, a change in magmatic flux that drives crustal thickening may be an important external forcing on climate and erosion. However, in the case of the dry Central Andes, the climate may also play a potentially important role as an external driver in the creation of a magmatic flare-up. In continental cordilleran arcs, ignimbrite volcanism often occurs within tectonically thickened and thermally softened crust[38]. The Central Andean ignimbrites are located in areas of high elevation and crustal thickness, and where precipitation is the lowest along the Andes[30]. The low precipitation arising from latitudinal variability in global circulation patterns along the Andes may cause a significant decrease in orogen-scale erosion rates, which can lead to crustal thickening and high elevations both from an increased change in storage in the mass balance of an orogenic wedge[48], as well as through starving the subduction zone trench of sediment and thus increasing the friction on the plate interface to levels that can support high topography[49]. It is therefore conceivable that climate itself may also be a factor in determining where along the arc flare-ups can occur.

We present a schematic cartoon illustrating our conceptual model for the growth of topography above the APMB since 10 Ma (Fig. 7), which builds off the models of Kay and Coira[5,13] and Beck et al.[7]. Beginning with the westward retreating delamination of a weak mantle lithosphere and dense lower crust after an increase in the dip of the subducting slab at 16 Ma (refs 5,7), hot (low density) asthenosphere flows under the previously thickened orogenic crust causing uplift from the gain in gravitational potential energy[50] (Fig. 7a). Decompression melting of the expanded mantle wedge and elevated temperatures at the base of the crust facilitate crustal melting and the addition of mantle-derived melts to the crust[5] (Fig. 4b), causing an additional Airy isostatic response (Fig. 7c).

Thus, in light of recent seismic constraints on the volume of partial melt in the mid-crust[15], we argue that magmatic thickening of the crust represents a significant component of surface uplift in the APVC since ~11 Ma. That the topographic response to building a batholith can be of similar magnitude to surface uplift from removal of the mantle lithosphere[26,27] has

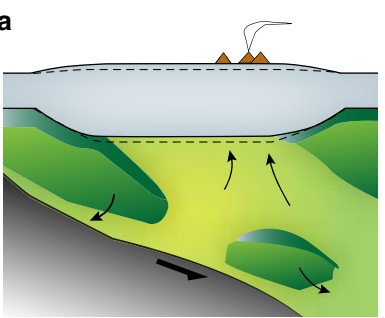 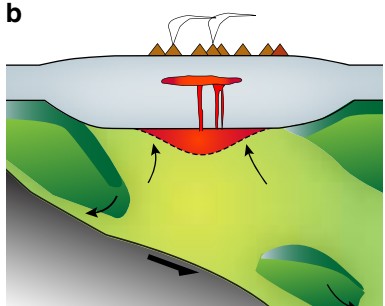 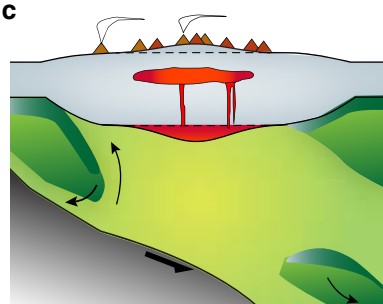

*16–10 Ma*: Retreating delamination and thermal isostatic response

*>~10 Ma*: Crustal thickening from intrusion of asthenospheric melt

*Today*: Airy isostatic response of thickened crust

**Figure 7 | Cartoon depicting the time evolution of topography and crustal structure along an east–west transect through the study area. (a)** Shows the retreating delamination and removal of the mantle lithosphere (medium green) and possibly lower crust (dark green) beneath the Puna plateau and consequent Pratt isostatic response to the emplacement of warm, lower density mantle asthenosphere (dunite green) beneath the crust. The subducting slab is shown in dark grey, and back-arc volcanism also persists during this time. **(b)** Depicts the influx of asthenospheric melt (reddish orange) into the crust (light grey) as an effective buried load. Airy isostatic compensation of this crustal loading from below leads to an increase in surface uplift in the Altiplano-Puna **(c)**.

implications both for interpreting the paleo-elevation history of the Central Andes as well as for geodynamical modellers seeking to understand the processes responsible for orogenesis and pluton growth in arc settings.

## Methods

**Topographic analysis.** To examine the surface topographic signal associated with the growth of the APMB, we utilize both the long wavelength topography and Paleozoic basement outcrop elevations generated from a 90 m SRTM digital elevation model. To look at the long wavelength component of the topography in our study area, we first downsample the SRTM data to a 1 km grid size, then take the two-dimensional Fourier transform of the data using Matlab's built in *fft2* command. We then multiply the transformed data set by a lowpass filter with a cutoff wavelength of 175 km, and finally take the inverse transform of the filtered data to recover the long-wavelength topography in *xy* space.

The basement rock exposures that dot the surface of the APVC consist of Paleozoic basement rock associated with the Antofalla terrane[19]. The folded and tilted strata associated with surface exposures of the Antofalla terrane near the APMB contrast markedly with the overlying deposits of volcanic rock, making their identification relatively straightforward in satellite imagery. We thus map exposures of this basement along the APMB using high-resolution satellite imagery in Google Earth, and extract their surface elevations using 90 m SRTM topographic data within ArcGIS. To visualize the data, we plot the outcrop centroid and its median elevation on Figs 2a and 3a.

**Isostatic modelling of melt contribution.** We use the long-wavelength topographic anomaly above the APMB to model the isostatic load at depth using a buried load isostatic model[24,25]. The conceptual model underlying the mathematical one assumes that the crust here is thickening from the addition of material at its base. In the case of our study, we assume that the crust is being added by mantle-derived melt as upwelling warm asthenosphere replaces the lithosphere that was removed through delamination[7]. We can therefore model the original contribution to crustal thickening from the accumulation of mantle-derived melt as a convolution of the topographic load profile and the isostatic response function, which is more simply solved through multiplication in frequency space:

$$W_{\mathrm{i}}(k) = -H_{\mathrm{b}}(k)\left(\frac{\rho_{\mathrm{c}}}{\phi_{\mathrm{e}}'''(k)\Delta\rho}\right) \quad (1)$$

Here $k$ corresponds to the wavenumber, $H_{\mathrm{b}}(k)$ is the Fourier transform of the topographic response profile, $W_{\mathrm{i}}(k)$ is the Fourier transform of the buried load profile, $\rho_{\mathrm{c}}$ is the density of the crust, $\Delta\rho = \rho_{\mathrm{m}} - \rho_{\mathrm{c}}$, and $\phi_{\mathrm{e}}'''(k)$ is the isostatic response function for loading at the base of the continental crust[25], defined as

$$\phi_{\mathrm{e}}'''(k) = \left(\frac{Dk^4}{\rho_{\mathrm{c}}g+1}\right)^{-1}. \quad (2)$$

$D$ is the flexural rigidity of the crust:

$$D = \frac{ET_{\mathrm{e}}^3}{12(1-v^2)}, \quad (3)$$

where $E$ is the Young's modulus, $T_{\mathrm{e}}$ is the effective elastic thickness of the crust and $v$ is Poisson's ratio (0.25). The parameters used in our calculations are found in

Supplementary Table 2. We then take the inverse transform of $W_{\mathrm{i}}(k)$ to find the initial buried load profile $W_{\mathrm{i}}$.

To mitigate the extreme sensitivity of the calculation to high-frequency noise, and isolate the component of long wavelength topography associated with the APMB, we approximate the long-wavelength topographic response profile $H_{\mathrm{b}}$ in one dimension as a Gaussian curve with an amplitude of ~1,200 m and a full-width at half-maximum of ~232 km (Fig. 4). This curve fits the long-wavelength topography well (Fig. 4); however, we model the isostatic load $W_{\mathrm{i}}$ for a range of dome amplitudes (Table 1). Solving Equation 1 using a dome amplitude of 1,200 m yields a buried load profile extending to about 9 km depth at its maximum (Fig. 4). To estimate the root volume from the one-dimensional profile $W_{\mathrm{i}}$, we assume radial symmetry of the load and calculate the volume $W_{\mathrm{iv}}$ numerically using trapezoidal integration.

We can relate our modelled isostatic root volume to mantle-derived melt volume utilizing an arc mantle production rate model[15,33]. As shown in Ward *et al.*[15] (their Supplementary Material), mantle-derived melt flux can be related to APMB volume via the following formulation:

$$\xi = \frac{\alpha(\beta+1)}{\left(\frac{\mu+1}{\eta+1}\right)\lambda\tau}. \quad (4)$$

Here, $\xi$ is the cross-sectional arc mantle magma production rate (km³ km⁻¹ per million years), $\alpha$ is the volume of volcanic material, $\beta$ is the plutonic:volcanic ratio, $\eta$ is the ratio of dense residue mass to melt mass, $\mu$ is the ratio of crust to mantle melt provenance, $\lambda$ is the arc-parallel length and $\tau$ is the time interval over which melt accumulation has occurred. As both the ratio of residue mass to melt mass ($\eta$) and crust to mantle provenance ($\mu$) are thought to be near 1:1 (ref. 13), the volume of mantle-derived melt calculated from the model should be equal to the seismically imaged volume of the magma chamber. This value is equivalent to the 'volume addition rate per arc length' as described by Paterson and Ducea[33].

To solve for the volume of the APMB, we relate our initial buried load isostatic volume $W_{\mathrm{iv}}$, to the cross-sectional growth of crust from magmatic thickening, $\xi$, by the following relationship:

$$\xi = \frac{W_{\mathrm{iv}}}{\lambda\tau}, \quad (5)$$

Plugging Equation 5 into Equation 4, and setting $\eta$ and $\mu$ equal to 1:1 as described above, allows us to calculate $\beta$ directly from topography and the volume of volcanic material $\alpha$.

$$\beta = \frac{W_{\mathrm{iv}} - \alpha}{\alpha} \quad (6)$$

Here $\alpha$ is approximately 15,000 km³ and represents the total volcanic volume (ignimbrites plus volcanic edifices). The APMB volume ($V_{\mathrm{apmb}}$) then is simply the volume of the total buried load $W_{\mathrm{iv}}$ minus the volcanic volume $\alpha$. Computer code for this analysis is available upon request to the corresponding author.

**Uplift history from the volcanic record.** We estimate the time history of surface uplift from magmatic thickening of the APVC crust by utilizing measurements of erupted ignimbrite volume (dense rock equivalent) since 11 Ma (Table 6 in Salisbury *et al.*[12]). Using our topographic estimate of $\beta = 32$, we convert ignimbrite volume to plutonic volume and divide the plutonic volume by the approximate magmatic footprint of the APMB (55,000 km²) to get a mean crustal thickness contribution. We convert the change in thickness to surface uplift assuming Airy isostatic compensation, and take the local gradient in uplift over time to estimate

surface uplift rate. As our buried load model shows, the large wavelength of the APMB does not feel the effects of the flexural rigidity of the upper crust and thus an Airy approximation for our simplified uplift model is appropriate. Because mechanical processes in the crust will filter the power input from the mantle[16], there is likely not a linear relationship between mantle melt production and surface eruption rates. However, the ~5 million years timescale that defines the peak APVC flare-up[12] may record the time variability of melt production[16], so we average our uplift calculations over this range of changes in volcanic volume.

**Data availability.** All the data for this manuscript are available upon request to the corresponding authors.

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

## Acknowledgements

Conversations with Emily Brodsky, Francis Nimmo and Sam Johnstone helped improve the quality of this manuscript. This work was financially supported by National Science Foundation grants EAR 0908850 (N.J.F) and EAR 1415914 (S.L.B., G.Z.).

## Author contributions

K.M.W., S.L.d.S., G.Z. and S.L.B. helped contribute the initial ideas that motivated this manuscript. J.P.P. performed the analysis with K.M.W. and input from all the authors.

S.L.d.S. analysed elevation distributions of the Altiplano-Puna plateau. K.M.W., G.Z. and S.L.B. developed the magmatic addition model, and J.P.P. wrote the manuscript with input from all the authors.

## Additional information

**Competing financial interests:** The authors declare no competing financial interests.

