## [Peer Review File · Nature Communications]

Reviewer #1 (Remarks to the Author):

This paper by Perkins et al regarding magmatically driven large magnitude uplift in the Altiplano-Puna plateau of the Andes orogenic system is a commendable piece of work. For a mountain system in which there are dozens of papers published (perhaps yearly, dozens) regarding the crustal thickening by fault-related orogenic activity and/or uplift by delamination or "drips", there have been few published since the 1970s on major contributions by magmatic input. The team whose years of careful work is combined in this submission, integrating deep knowledge of the magma system and deep knowledge of the overall lithospheric properties, have succeeded in a well constrained new view of the magmatic system.

Although the references to the Sierra Nevada magmatic arc system of western North America are so brief as to probably be inadequate, the inclusion of that comparison is important. Deeply eroded orogenic systems expose large volume batholiths. This study connects those batholiths to a modern mountain system and illuminates the surface topographic change that we ought to anticipate accompanied the production of those ancient batholiths.

While the data and analysis are excellent, I find the paper to be in need of careful editing. The authors seem too close to the topic, and have not explained the logic of their approach to non-experts. There are some places in the text that the explanations are vague, and places that the text explanations of the figures are inadequate or confusing. Labels of the figures are inadequate in some basic ways. Two or more items are both defined as "beta", or a single "beta" is described in two such different manners that it seems like there must be two items. All these points are made in "comments" now embedded in the copy of the pdf file that I edited.

Overall, I think this paper is worthy of publication in Nature Geoscience. But I think it needs editorial work.

Reviewer #2 (Remarks to the Author):

This study takes an innovative new approach to evaluating the magnitude of surface uplift related to magmatic addition to the crustal lithosphere by arc magma production. Through mapping of basement exposures and evaluation of modern topography, the authors show that a 1 km high topographic dome is associated with the seismically imaged extent of the Altiplano-Puna magma body. Using a buried load isostatic model, the authors estimate an arc-mantle magmatic production rate and demonstrate that the rates of surface uplift associated with magmatic addition to the crust may be comparable with surface uplift rates associated with lower lithosphere removal. The methods used in the analysis are robust, and the authors use modest estimates in calculating the magma production rates. The surface uplift rates may be represented at a slightly higher rate than their analysis suggests, but this can be easily addressed (see below). The paper is very clear and concise; the figures are relevant to the study and clear; and the references to prior work are appropriate. This study provides an unprecedented view of the 'significant' role of magmatic addition to the growth and surface uplift of mountain belts. It will be of broad interest to the tectonics community. I would encourage publication of this manuscript with minor revisions. Below I have provided specific suggestions/comments keyed to the text.

Comments:

Figure 1. In either Figure 1a or 1b, show the location of the cross sections represented in figure 2.

Line 56-58. I don't see the dashed line referenced here in figure 2A. Please clarify the text or correct the figure.

Lines 89-90. Are the mean elevations for the mapped basement outcrops shown as the blue dots in Figure 2A? If so, then please make that clear here or in the figure caption.

Line 107. It is not clear from figure 2 why the authors use a 1200 m amplitude. The amplitude appears to be ~900 m if comparing the southern Puna to the APVC dome and is ~1500 m if comparing the southern Altiplano to the APVC. Are the authors somehow splitting the difference between north and south of the dome? Please add a brief explanation to make this clear.

Line 133-135. This number is a little higher than what is shown in Figure 3. Based on Figure 3, the maximum rate achieved between ~6 and 5 Ma appears to be 0.22 mm/year. For a more accurate representation of the figure and the analysis, I would suggest revising this sentence to "Maximum estimated surface uplift velocity over the flare-up is approximately 0.22 mm/yr between ~6 and 5Ma, . . . "

Line 191. Again, please explain why an amplitude of 1200 m is used.

Lines 336-337 (figure 2 caption). Are the "median outcrop elevations (blue dots)" referenced here the 'basement' outcrops discussed in the text? If so, then please add 'basement' before "outcrop" here to make this clear.

Line 344 (figure 3 caption. Correct the superscript "??" after Ma.

Line 346. There appears to be another problem here with the superscript after "input".

In the Figure caption for Supplementary Figure 3 "ascending is misspelled".

In the caption for Supplementary Tabl

Reviewer #3 (Remarks to the Author):

I enjoyed reading this manuscript.

Most importantly, this manuscript proposes that magmatic thickening plays a fundamental role in uplift. Although I have not followed all the literature on the Andes, it is my impression that most think that the thickness of the Andean crust (and by implication elevation) is primarily controlled by shortening. However, I have always myself been skeptical of this, mainly because constraints on magmatic thickening have in the past been so very difficult to determine. Thus, it is encouraging that these authors have found evidence that magmatic thickening is important. I believe that this work is timely, robust and will generate new discussion on the origin and evolution of mountain belts.

My comments below are mainly to help the manuscript improve, if necessary.

I very much like their analysis of the gravity. It is refreshing to see someone use the free-air gravity anomaly to show, as it should almost always be, that elevations are isostatically compensated at least on long wavelengths. The Bouger analysis is also nice. What would be also helpful is if they could plot the Moho depth as a function of latitude as well. Is this not known well in that area? It would seem that a plot of elevation versus Moho thickness would seal the deal that the elevations are compensated primarily by a crustal root. Why not show this?

I also think the authors have missed an opportunity to really think about the bigger picture. They have already identified that magmatism plays a role in uplift. Plotting uplift rates with magmatic inputs is a great idea. But if there is uplift, there is also erosion, so the authors may want to add a

few sentences thinking about the feedbacks between erosion, uplift and magmatism. The authors may wish to look at this paper, where we discuss the importance of magmatism on uplift and crustal thickness:

Lee, C-T A, Thurner, S., Paterson, S., Cao, W., 2015, The rise and fall of continental arcs: interplays between magmatism, uplift, weathering, and climate, Earth and Planetary Science Letters, doi: 10.1016/j.epsl.2015.05.045.

Note that the magmatic fluxes on our observation figure is wrong (it is right in our models) - this was Scott Paterson's fault. I had originally had the right number based on my own calculations, but Paterson edited the figure and I unfortunately did not catch the error. I bring this to your attention because his numbers in the Elements article that you cite are also wrong. The numbers in Ducea's GSA Today paper are also wrong. By wrong, I mean that there's an order of magnitude typo or wrong units - you should contact them to be sure. So just make sure you're comparing to good numbers for the sierras.

Dear Editor Plail,

Please find below our line-by-line responses to reviewer and editor comments. Reviewer and Editor comments are in italicized type, and author responses are in bold.

Editor Plail

As your manuscript was transferred from Nature Geoscience, I would like to just make you aware that Nature Communications in fact permits articles to be up to 5000 words in length, plus an unlimited Methods section. We permit up to 10 display items (i.e. figures and tables), which may be accompanied by captions up to 350 words in length. We also permit up to 70 references.

We have now folded in the supplementary figures and tables to the main text, and expanded our results, discussion, and methods section to be more in line with the style of Nature Communications.

Reviewer #1 (Remarks to the Author):

This paper by Perkins et al regarding magmatically driven large magnitude uplift in the Altiplano-Puna plateau of the Andes orogenic system is a commendable piece of work. For a mountain system in which there are dozens of papers published (perhaps yearly, dozens) regarding the crustal thickening by fault-related orogenic activity and/or uplift by delamination or "drips", there have been few published since the 1970s on major contributions by magmatic input. The team whose years of careful work is combined in this submission, integrating deep knowledge of the magma system and deep knowledge of the overall lithospheric properties, have succeeded in a well constrained new view of the magmatic system.

Although the references to the Sierra Nevada magmatic arc system of western North America are so brief as to probably be inadequate, the inclusion of that comparison is important. Deeply eroded orogenic systems expose large volume batholiths. This study connects those batholiths to a modern mountain system and illuminates the surface topographic change that we ought to anticipate accompanied the production of those ancient batholiths.

While the data and analysis are excellent, I find the paper to be in need of careful editing. The authors seem too close to the topic, and have not explained the logic of their approach to non-experts. There are some places in the text that the explanations are vague, and places that the text explanations of the figures are inadequate or confusing. Labels of the figures are inadequate in some basic ways. Two or more items are both defined as "beta", or a single "beta" is described in two such different manners that it seems like there must be two items. All these points are made in "comments" now embedded in the copy of the pdf file that I edited.

Overall, I think this paper is worthy of publication in Nature Geoscience. But I think it needs editorial work.

Line-by-line comments and responses:

1. *The text may be easily followed by a specialist, but a well-informed non-specialist found it hard to follow the threads of logic. Perhaps as a consequence of there being no section titles, I found the flow of interconnected ideas through the text to be unclear. Individual paragraphs are clear, but there needs to be more effort to explain the connections among the multiple steps in the analysis. Early in the paper, tell us where are we going, and how will the pieces will get us there.*

This point is well-taken, and in an effort to clarify the threads in our paper we have added section headings in the Results section of the paper as well as an additional paragraph at the end of our introduction that provides more of a road map for how the rest of the paper will continue.

47. *I do not see how this paper clarified whether melt intrusion is "reflected in the geology". Topography, yes. Geology, no.*

This was poor wording on my part, and I have now removed this reference to the geology of the APVC.

58. *Your reader cannot follow your intent in these several sentences. We see in Fig. 2A a cross section with some short-wavelength lows and some short wavelength highs. We do not know if the "summits of the late Miocene volcanoes" have something to do with those spiky high elevations or not. We see only topography, not lithology or age. Maybe those high points are Quaternary volcanoes. If so, where on the diagram do I see Miocene volcanoes? How would I know whether the slightly broader high zones in the south are Miocene volcanoes or fault blocks? Your figure does not convey this information.*

Since this information was extraneous and did not add anything more to our analysis, we've gotten rid of this section in the results paragraph and removed the hand-drawn envelope from our swath topographic profile above the APMB in Figure 2 (mentioned below).

58. *Not sure how "this is akin to looking at the long wavelength component of the raw topography" differs from (or complements) lines 56-57 statement "...on the long wavelength topographic dome..." Is that envelope calculated using some particular averaging scheme? Or is it a free-hand approximation?*

We have removed these lines from the text now, as this discussion was not particularly clear. In its place there is now a paragraph discussing the amplitude of the long-wavelength dome (in response to comments from Reviewer #2).

68. *Not yet clear if and why it matters that the lithosphere is thin because of that specific mechanism. Gravity and topography inform us about static conditions. If other removal mechanisms would not fit the static conditions, then you need to introduce this argument directly, not hint at it.*

We agree with Reviewer #1's point that this sentence appears out of place. I have now moved discussion of the mantle lithosphere to our Discussion section where we place our observations in the broader context of Andean tectonics and surface uplift.

70. Neither Figure 1 nor Figure 2 gives the reader any clue as to where the Altiplano is located, or southern Peru. Do the authors intend that "southern Altiplano" be understood to be north of the APMB, which is in the Puna? Or do they use "altiplano" to be inclusive of the entire Altiplano and Puna. Vague.

In order to clarify the locations of the Altiplano and Puna plateaus, we have labeled both Figures 1 and 2 accordingly. Additionally, we have changed the language of this sentence to now say, "...although there is a nonzero free-air gravity anomaly along the Central Andes, it is low and does not change markedly from the Altiplano to the Puna (Fig. 2c)."

70. omit "the"

Corrected.

90. "exposures in general decline" is confusing. They decline in elevation? Or they are progressively less common?

To make this sentence read better, we have now changed the wording to "Although exposures of basement rock become less common toward the interior of the APVC....."

98. This first sentence is key to understanding the next major analysis, yet it is not clear. Reader gathers that they assume isostatic compensation. The second phrase (connecting topography, crustal thickness, mantle mass addition, and gravity) is NOT clear. It should be stated in logical steps.

We have re-tooled this description of our model results in the section "Modeling melt production from topographic data." We now start this paragraph with the following sentences: "The presence of the large-volume APMB implies a substantial input of mantle-derived magma into the crust of the Altiplano Puna Volcanic Complex. As the regional gravity data suggest that the topography of the APVC is in isostatic equilibrium, we can therefore exploit the topographic high above the APMB to learn about the contribution of magmatic thickening to surface uplift.

102. Not clear why an assumption is stated that is not correct, and because it is not correct two different densities are assumed. If these two densities were to be assumed, then why was it first stated that the new material is of the same density as the crust. If authors are trying to confuse the non-expert reader, they are doing well.

We recognize that this was a problem of wording regarding the necessary simplifications needed in our isostatic model, so we have now adjusted the language in the second paragraph of our isostatic model Results subsection. The sentence now states, "Our isostatic

model requires a constant crustal density, so we take an average density of 2800 kg/m³ for the crustal column and 3250 kg/m³ for the density of the underlying asthenosphere.”

120. a little confusing. Since the methods section may not have been read, why use the term "beta" when "ratio" is more self explanatory, ie., ..."calculate plutonic:volcanic ratio of 38 and an ..."

I've now changed this sentence to read, “..we calculate a plutonic:volcanic ratio (β) of 38 and an arc mantle magma production rate (ξ) of ~200 km³/km/Myr for the APMB (Equation 5, Table 2; see *Methods* section for details).” This way, hopefully the calculated values make more sense to a reader who has not yet read the methods section, yet still provide a reference to the equations used.

124. line 204 (methods) states that beta equals the plutonic:volcanic ratio. Why then does this imply (repeated in lines 2014-215) that beta equals a "topographic estimate"? Very confusing.

Here our goal was to compare our modeled result of the plutonic:volcanic ratio to other batholith systems. Since the information our model uses to calculate plutonic volume is primarily the topography, we referred to this value as “our topographic estimate of β .” This clearly was confusing, so we have now changed the sentence to read, “Additionally, our topographically constrained estimate of the plutonic:volcanic ratio ($\beta=38$) for the APMB is similar to geochemical estimates of $\beta=30$ for the Sierran batholith.”

127. need to be specific about the time span referred to here. Total uplift since 11 Ma? Total uplift since the Cretaceous?

We've now changed the sentence to read, “The ~1 km of surface uplift gained over the 11 Myr of magmatic addition represents ~1/5 of the total modern-day elevation of the Altiplano-Puna, with the remainder of surface uplift being accomplished primarily through crustal shortening^{10,4} and lithospheric removal¹³.”

132. add "the"

Corrected.

140. Time sequence is unclear. This sentence implies that the "thickened orogenic crust" needs compensation, but is unclear regarding whether this crust thickened in the 16-10 Ma time span, or earlier. Figure 4a shows nothing that looks like "orogenic thickening." Reader thinks "orogenic thickening" will look like it is related to horizontal shortening. If it means magmatic addition (visible in Figure 4b) then the wording could better point to the magma. Yet the next sentence clearly is intended to cover Figure 4b, leaving reader looking for orogenic thickening in Fig. 4a but finding none.

Here we are attempting to illustrate our conceptual model of magmatic thickening *after* an initial phase of orogenic thickening and subsequent lithospheric removal at the latitudes of the APVC. In order to make this more clear, we have now changed the sentence to say,

“Beginning with the westward retreating delamination of a weak mantle lithosphere and dense lower crust after an increase in the dip of the subducting slab at 16 Ma¹³, hot (low density) asthenosphere flows under the previously thickened orogenic crust causing uplift from the gain in gravitational potential energy.”

143. *add of*

Corrected.

Reviewer #2 (Remarks to the Author):

This study takes an innovative new approach to evaluating the magnitude of surface uplift related to magmatic addition to the crustal lithosphere by arc magma production. Through mapping of basement exposures and evaluation of modern topography, the authors show that a 1 km high topographic dome is associated with the seismically imaged extent of the Altiplano-Puna magma body. Using a buried load isostatic model, the authors estimate an arc-mantle magmatic production rate and demonstrate that the rates of surface uplift associated with magmatic addition to the crust may be comparable with surface uplift rates associated with lower lithosphere removal. The methods used in the analysis are robust, and the authors use modest estimates in calculating the magma production rates. The surface uplift rates may be represented at a slightly higher rate than their analysis suggests, but this can be easily addressed (see below). The paper is very clear and concise; the figures are relevant to the study and clear; and the references to prior work are appropriate. This study provides an unprecedented view of the 'significant' role of magmatic addition to the growth and surface uplift of mountain belts. It will be of broad interest to the tectonics community. I would encourage publication of this manuscript with minor revisions. Below I have provided specific suggestions/comments keyed to the text.

Comments:

Figure 1. In either Figure 1a or 1b, show the location of the cross sections represented in figure 2.

We've now denoted cross section lines in Figure 1b as thin yellow lines, and added this description in the figure caption.

Line 56-58. I don't see the dashed line referenced here in figure 2A. Please clarify the text or correct the figure.

This was an editorial error on my part in reference to a section of Figure 2a that no longer exists. We have changed the text in this paragraph accordingly.

Lines 89-90. Are the mean elevations for the mapped basement outcrops shown as the blue dots in Figure 2A? If so, then please make that clear here or in the figure caption.

We've now added a description of the blue dots in the Figure 2 caption. We mention the dots a few lines down in the text, so have kept the language the same in the main part of the document.

Line 107. It is not clear from figure 2 why the authors use a 1200 m amplitude. The amplitude appears to be ~900 m if comparing the southern Puna to the APVC dome and is ~1500 m if comparing the southern Altiplano to the APVC. Are the authors somehow splitting the difference between north and south of the dome? Please add a brief explanation to make this clear.

Indeed we are splitting the difference, since the APVC exists at the boundary between two physiographic provinces with differing mean elevations. In the second paragraph within the first Results subsection (entitled "Isolating the topographic signature of the Altiplano-Puna Magma Body"), we now include a discussion on quantifying the amplitude of the topographic dome above the APMB.

Line 133-135. This number is a little higher than what is shown in Figure 3. Based on Figure 3, the maximum rate achieved between ~6 and 5 Ma appears to be 0.22 mm/year. For a more accurate representation of the figure and the analysis, I would suggest revising this sentence to "Maximum estimated surface uplift velocity over the flare-up is approximately 0.22 mm/yr between ~6 and 5Ma, . . . "

We've corrected this sentence to read, "Estimated surface uplift velocity over the flare-up is between 0.20 and 0.25 mm/yr..." The range represents hopefully a more truthful estimate of geologic uncertainty.

Line 191. Again, please explain why an amplitude of 1200 m is used.

In this *Methods* subsection we now explain our choice of using a 1200 m amplitude approximation as our best characterization, and state that we model a range of dome amplitude values within our estimated range (which are presented in Table 2).

Lines 336-337 (figure 2 caption). Are the "median outcrop elevations (blue dots)" referenced here the 'basement' outcrops discussed in the text? If so, then please add 'basement' before "outcrop" here to make this clear.

Corrected.

Line 344 (figure 3 caption. Correct the superscript "???" after Ma.

Line 346. There appears to be another problem here with the superscript after "input".

Corrected. We have also modified this figure caption to more appropriately describe the figure in its present form.

In the Figure caption for Supplementary Figure 3 "ascending is misspelled".

Corrected.

Table 1, correct the CIT? superscript.

Corrected.

Reviewer #3 (Remarks to the Author):

I enjoyed reading this manuscript.

Most importantly, this manuscript proposes that magmatic thickening plays a fundamental role in uplift. Although I have not followed all the literature on the Andes, it is my impression that most think that the thickness of the Andean crust (and by implication elevation) is primarily controlled by shortening. However, I have always myself been skeptical of this, mainly because constraints on magmatic thickening have in the past been so very difficult to determine. Thus, it is encouraging that these authors have found evidence that magmatic thickening is important. I believe that this work is timely, robust and will generate new discussion on the origin and evolution of mountain belts.

My comments below are mainly to help the manuscript improve, if necessary.

I very much like their analysis of the gravity. It is refreshing to see someone use the free-air gravity anomaly to show, as it should almost always be, that elevations are isostatically compensated at least on long wavelengths. The Bouger analysis is also nice. What would be also helpful is if they could plot the Moho depth as a function of latitude as well. Is this not known well in that area? It would seem that a plot of elevation versus Moho thickness would seal the deal that the elevations are compensated primarily by a crustal root. Why not show this?

Reviewer #3 makes an excellent point that a plot of moho depth with latitude would be a helpful graphic. However, we avoid appealing to previous estimates of crustal thickness for the following reasons. 1) Regional variability in upper mantle structure changes the relationship between crustal thickness and elevation from the Northern Altiplano to the Southern Puna (e.g., Beck et al., 2014; Ward et al., 2015 AGU abstract). This complicates any plot of crustal thickness vs. latitude for the Central Andes, which I believe may distract from our focus solely on the APVC. 2) Melt segregation processes that lead to batholith growth leave dense cumulates at the base of the crust and thus create a broad “transition zone” between the lower crust and the upper mantle. The seismic velocities of cumulates in the lower crust may make distinguishing the seismic Moho difficult (e.g., Lipman and Bachman, 2015), and is not incorporated in crustal structure models of the Andes (e.g., Tassara and Echaurren, 2012). Because of these complications, we appeal largely to the unchanging free-air gravity anomaly across the long-wavelength dome as evidence that the excess mass in the high topography of the APVC is compensated at depth.

I also think the authors have missed an opportunity to really think about the bigger picture. They have already identified that magmatism plays a role in uplift. Plotting uplift rates with magmatic inputs is a great idea. But if there is uplift, there is also erosion, so the authors may want to add a few sentences thinking about the feedbacks between erosion, uplift and magmatism. The authors may wish to look at this paper, where we discuss the importance of magmatism on uplift and

crustal thickness:

Lee, C-T A, Thurner, S., Paterson, S., Cao, W., 2015, The rise and fall of continental arcs: interplays between magmatism, uplift, weathering, and climate, Earth and Planetary Science Letters, doi: 10.1016/j.epsl.2015.05.045.

We thank Reviewer Lee for bringing our attention to this paper – I had not seen it previously and it is very relevant to our manuscript! We have now included an enhanced discussion placing the APVC in the context of this recent modeling work on magmatic addition, crustal thickness, weathering/erosion, and climate.

Note that the magmatic fluxes on our observation figure is wrong (it is right in our models) - this was Scott Paterson's fault. I had originally had the right number based on my own calculations, but Paterson edited the figure and I unfortunately did not catch the error. I bring this to your attention because his numbers in the Elements article that you cite are also wrong. The numbers in Ducea's GSA Today paper are also wrong. By wrong, I mean that there's an order of magnitude typo or wrong units - you should contact them to be sure. So just make sure you're comparing to good numbers for the sierras.

Author KMW spoke with Scott Paterson over email, and we received a revised figure from him with appropriate MARs for the Sierran Batholith that we have included in our revised manuscript.

Cin-Ty